# A 24-year longitudinal study on a STEM gateway general chemistry course and the reduction of achievement disparities

**Partha Basu**[1]*, **David J. Malik**[1]*, **Steven Graunke**[2]

1 Department of Chemistry and Chemical Biology, Indiana University Indianapolis, Indianapolis, Indiana, United States of America, 2 Analysis and Institutional Effectiveness, Indiana University Indianapolis, Indianapolis, Indiana, United States of America

* basup@iu.edu (PB); dmalik@iu.edu (DJM)

## Abstract

The "First Year Experience" is a critical component of retention of STEM majors. Often, general chemistry has been labeled as a "gatekeeper" course for STEM careers due to a high attrition rate and a course that leads to increased time for graduation when students are inadequately prepared. We demonstrate that the active learning strategy *Peer-Led Team Learning* (PLTL) model increases student retention (%DFW calculated from earned grades A through F plus withdrawals, W) and success (%ABC calculated from earned grades A through F). We have analyzed approximately 24 years of data in general chemistry I (~20,000 students), using Analysis of Covariance (ANCOVA), which showed progressive, significant improvement in both student success and completion metrics. A Hierarchical Linear Modeling (HLM), using a combination of course and student-level variables, demonstrated the impact of PLTL on internal exam metrics and overall course grades. Further, HLM modeling assessed the impact of PLTL controlling for various student demographics. PLTL strongly impacted URM student completion rates to a greater degree than well-represented students, reducing the URM/non-URM achievement gap.

## Introduction

The importance of Science, Technology, Engineering, and Mathematics (STEM) towards the progress of society has long been emphasized, in fact, it dates back to 1945 when Vannevar Bush articulated the need in a letter to the president [1]. Since the crisis of Sputnik in 1957, it has been emphasized that the United States is losing its standing in the international leadership in technology due to the weakness in educational preparedness in STEM [2–4]. More than 65 years later, the problem still persists. In 2009, the President's Council of Advisors on Science and Technology announced that if the United States is to retain its preeminence in science and technology and remain competitive, it will need one million more STEM professionals over the next decade than it is currently projected to produce [5]. Post-secondary education plays a critical role in building a strong STEM workforce, and national data revealed that more than half of first-year students who declared STEM majors at the start of college left the field before graduation. More than half of STEM bachelor degree recipients switched to non-STEM fields when entering the Graduate School or labor market [6]. Students with

**Data availability statement:** All relevant data are within the manuscript and its Supporting Information files.

**Funding:** The author(s) received no specific funding for this work.

**Competing interests:** The authors have declared that no competing interests exist.

weaker backgrounds leave STEM fields more frequently than others, and it takes longer for those students to graduate in STEM disciplines. Interestingly, while about 28% of first-year students expressed an interest in STEM disciplines, actual enrollment is much lower (∼14%). The report noted that about 56% of post-secondary students who declared STEM majors in their first year left these fields over the next six years. Studies have shown that women, underrepresented minorities (i.e., Hispanic, Black, American Indian, Alaska Native, or Native Hawaiian/Pacific Islander), first-generation students, and those from low-income backgrounds based on Pell grant eligibility exit STEM fields at higher rates than their counterparts. Non-URM students in our study include those who self-identify as white or Asian, or do not identify any race. URM students usually start college with the same level of interest in STEM majors as their non-URM peers, but six-year STEM completion rates drop from 52% for Asian Americans and 43% for White to 29% for Hispanic/Latine, to 25% for Native Americans, and to 22% for Black students [7,8]. Disparities in STEM degree attainment are also pronounced for low-income versus higher-income students [9]. It has been suggested that negative experiences and poor performance in gatekeeper or introductory math or science courses often contribute to attrition from STEM disciplines [10]. Negative experiences may include large class sizes, passive learning techniques, and lack of direct contact with faculty members. In addition, language may play a role for international instructors and students [6]. Bensimon argued that the achievement gap between well-represented and underrepresented students is one of higher education's most urgent and intractable problems [11]. The term "achievement gap" has received attention as a less appropriate term emphasizing deficit thinking. Our use employs the term as a comparative metric and that our efforts are to demonstrate appropriate pedagogies that refocus attention to improving obstacles to learning [12]. Given the role of General Chemistry for all STEM majors, it is critical to address factors that can improve student retention and success in this essential course [13–17].

The General Chemistry course, typically taught in a two-semester sequence in the first year [18,19], has been described as one such gatekeeper course. In the first semester, the primary focus is on electronic structure, thermochemistry, and stoichiometry, and in the second semester, the primary focus is on thermodynamics and kinetics. Without a meaningful prior chemistry background, these concepts are challenging for many students. Usually, a one-year high school chemistry course (or a one-semester college introductory chemistry class) is required. How to make the General Chemistry course more successful has been debated for decades. Indeed, Havighurst raised a call to reform general chemistry in 1929 [20]. In many cases, traditional lecture methods are augmented with recitation, where students get help from trained teaching assistants, often graduate students. Because general chemistry is a multi-semester foundational course in STEM disciplines, virtually all science and engineering programs require students to have a solid background in general chemistry. The content has a significant role in preparing students for careers in medicine, dentistry, pharmacy, science, and engineering. Strong academic performance and experiences in general chemistry can improve self-efficacy, which in turn sets the students onto a more satisfying career trajectory [21–23].

Active learning strategies have improved student outcomes in various educational settings. These strategies include think-pair-share, problem-based learning (PBL), flipped classes, case-based learning, interactive lectures, and forms of peer involvement. In chemistry classes, in-class demonstrations and computer-generated models can also engage students. A common theme in these strategies is the improvement of student engagement with the content to develop better problem-solving skills and enhance higher-order thinking. It is generally accepted that when students are engaged in group discussions and collaborative work their overall understanding improves. A meta-analysis has reported that active learning alone increases exam performance by 6% on average, and students in the traditional lecture format

are 1.5 times more likely to fail than those in active learning settings [24]. Despite the known outcomes of these strategies, they are not widely accepted. There are several potential reasons for the reluctance, which include lack of institutional support in setting up active learning classrooms and reduction of class sizes, reluctance from the faculty to adopt new approaches from traditional lecture styles, and support from the students [25]. Regardless, among these strategies, peer engagement in learning, such as Peer-Led Team Learning (PLTL), has a special space as the learning process is primarily driven by undergraduate students helping guide and shape better engagement with course content by bridging the gap between the content expert (faculty) and the student learner interfaces. The PLTL paradigm [26] trains recent course completors in strategies to engage students in the learning process. They meet in weekly small student groups (workshops) to solve current problems using a course workbook synchronized to topical progression in the course. The recent completors (called "leaders") guide the students to collaboratively explore solving these problems, not as tutors, but as facilitators to help the students discover strategies in problem-solving. Specific examples of a session can be found in the workbook [27]. The workbook units contain three different parts: (*i*) a self-test to prepare students using prior content knowledge to solve new problems (leveling), (*ii*) current collaborative group problem-solving exercises, and (*iii*) post-workshop exercises to allow students to independently exploit and reinforce learning strategies gained from the small group meetings. The workshop and workbook approach is based on the work of Vygotsky and his Zone of Proximal Development [28] where students first level their knowledge to a common basis, then new learning is moderated to transition the student to higher levels of understanding via a *More Knowledgeable Other* (leader), followed by a final phase of strengthening the acquisition of new knowledge (via reinforcement in the post-workshop exercise) and in preparation for the next workshop.

Since the Fall 1998 term, the Department of Chemistry and Chemical Biology at Indiana University Indianapolis (previously known as Indiana University-Purdue University Indianapolis, IUPUI) has implemented PLTL. Over the years, several articles on the efficacy, outcomes, and structure of PLTL have been published [13,23,29–33].

To better understand the impact of PLTL pedagogy on previously reported student success [22,34–36], we have collected data from 1996 through 2019 on 19,911 students. With our decades-long longitudinal data, we ask what the hallmarks of this pedagogy have been and whether or not the early, favorable metrics have been sustainable. The objectives of this study are to: (i) assess the long-term trends with respect to early success/completion metrics and controlling for other student-centric factors, (ii) determine how evolving student demographics impacted PLTL efficacy and effectiveness, and (iii) determine the roles of internal course metrics and engagement assessments on outcomes and potential grade inflation.

We limit the discussion to the structure and assessment in the context of our institutional data, not to the principles of PLTL. Analysis of Covariance (ANCOVA) was used at the course level to determine if %ABC and/or %DFW significantly improved after adopting PLTL pedagogy. Hierarchical Linear Modeling (HLM) included a combination of course and student-level variables to determine the impact of engagement in PLTL on individual in-course assessments. Further HLM modeling assessed the impact of PLTL on URM students and Pell grant recipients among other factors.

## Materials and methods

### General approach to PLTL at our institution

Embracing the critical components of a PLTL program [23], the Department strived to address the following in its implementation: (i) organizational structure to promote learning

limited to six to eight students per group; (ii) workshop activities emphasize active learning problems designed for collaborative learning, parsing problem solutions to multiple students, and careful integration with lecture curriculum and sequencing; (iii) peer leaders, recent successful course completors, are continually trained and supervised; (iv) faculty teaching the course are involved with PLTL components including selection and training of leaders; (v) PLTL session attendance is mandatory and participation is essential, and students receive the credit that reflects the extent of their engagement with the process; (vi) institutional support for innovative teaching and high impact practices, and (vii) students learn why the course has a PLTL component, how to work in a team, and receive evidence about the efficacy and benefits of PLTL.

An important feature of the program involved designing a workbook for use by students that establishes an expectation of preparation for the weekly meeting, problems during the workshop tuned to optimize active learning principles, and a follow-up set of exercises that reflect expectations of learning. Details are included in the supplementary information.

Student readiness for the course is essential. Many institutions use placement examinations and/or pre-requisite courses to ensure the student is familiar with the standard of a year of high school chemistry. To assess their readiness for general chemistry, we require students to take a Chemistry Placement Examination (CPE) designed by our instructors. Over the period of our study, we have also used the "Assessment and LEarning in Knowledge Spaces" (ALEKS) math placement exam [37,38] to confirm suitable co-requisite preparation. Those whose chemistry competency level is less than adequate are advised to complete a one-semester introductory chemistry course.

An analysis of the 2017 fall offering requiring both the CPE and ALEKS exams for enrollment guidance revealed decreasing Pearson correlations ($p < .0001$) of the CPE scores to the first three exams of 0.38, 0.37, and 0.25, respectively, and representing 15% to 6% contribution of variability in scores from the CPE. Similarly, for the ALEKS exam, correlations were 0.26, 0.23, and 0.19, representing 7% to 4% of the variability of exam scores. On the total course points, CPE and ALEKS correlations are 0.32 vs. 0.24, respectively, suggesting that 10% (at best) of the variability of those points might only be of modest usefulness in predicting course success, yet slightly better at informing course readiness that predicts 15% (*vide infra*).

## Limitations

Like other studies, this study also has limitations. Some data may be missing for various reasons, such as data loss due to a change in the Student Information System (SIS), non-required SAT scores with transfer students, and some high school and international students' GPAs missing. Some students may have dropped the course before exams 3 and 4 and could not be included in the HLM analysis. It is important to note that all analyses (ANCOVA, HLM, and OLS) were conducted using only students with a value for every measure in the analysis. The race and gender data in this study came from self-identification and may not reflect the true case. Finally, the data were compiled at one large, urban-serving institution in the Midwest, and institutions with different student populations may experience different results. However, our central finding is that when applied correctly, PLTL is an effective intervention tool for student success.

## Data collection and statistical analysis

The Department of Chemistry and Chemical Biology implemented the PLTL approach in general chemistry I, Chem C105, in the fall of 1998. This course is taught in fall, spring, and summer terms. The fall semester is considered the "regular" offering, while the spring

semester is the "off" semester. Data from two years prior (1093 students; 15% URM) to the start of the PLTL program serve as the control. Fall 2019 was selected as the final term in this study to account for the move to mostly remote instruction during the Spring 2020 semester due to COVID-induced disruptions, both in the offering and students' preparedness. Thus, we have analyzed data from 19,911 students (17% of them are URM) in courses completing C105 between Fall 1996 and Fall 2019. Data collection followed the protocol titled "*Efficacy of Placement Exam Data on Predicting Course Success,*" approved by the Indiana University Institutional Review Board (IRB) as an exempt protocol (study number 1801028552) dated February 13, 2018, with no expiration date.

**Student success and completion analysis.** In the first case, we determined the %DFW and the %ABC for each semester as metrics of student completion and success, respectively. Presumably, the assignment of an A, B, or C grade (any flavor of grade A + to C-) indicates the student met the requirements for a course to count toward a major. We surmise that a higher value of the percent ABC grades reflects a favorable outcome. For assignment of a D, F, or W (reflecting withdrawal from the course prior to term completion) grade, a lower %DFW value is considered more favorable for course assessment and as a measure reflecting student completion. Overall, this is a measure of unacceptable completion since either D or F grades usually disqualify the course from counting towards a student's degree requirements. The outcomes of %DFW and %ABC rate changes are discussed in the results section.

**Analysis of covariance at the class level.** The second analysis used is the Analysis of Covariance (ANCOVA) to determine if significant differences exist between courses before PLTL implementation and those with PLTL and the effect of the academic and demographic characteristics of students enrolled in each course. Individual student demographic data was obtained from the university student information system and merged with course grades. These demographic data were then aggregated at the course level for these analyses. Demographic and academic data included the percentage of URM students, percentage of students identifying as female, percentage of students indicating they were first generation (as indicated in their admission application), percentage of students receiving a Pell grant, and total SAT score. Because of a change in the Student Information System in 2003, incoming GPA was not found for all students in many of the older years in the 1990's, which is why SAT score was used as a proxy of incoming academic ability. In addition, term differentiation (Fall vs. Spring) was also included in the model to account for differences in course-level performance by semester. Finally, placement test differentiation via three scenarios: The ACS Toledo Examination, ALEKS Math only, or in-house Chemistry Placement Exam (CPE) with the ALEKS Math. The ACS exam was abandoned because it could not be administered online. Using the ALEKS-only exam, the DFW grades increased slightly. Most recently, we have used the CPE and ALEKS together scores for placement recommendations.

The ANCOVA theoretical model depicted in Eq. 1 is a linear combination of several factors:

$$Y_i = \mu + \tau_i + \beta_1\left(x_{1i} - \overline{x_1}\right) + \beta_2\left(x_{2i} - \overline{x_2}\right) + \beta_3\left(x_{3i} - \overline{x_3}\right) + \beta_4\left(x_{4i} - \overline{x_4}\right)$$
$$+ \beta_5\left(x_{5i} - \overline{x_5}\right) + \beta_6\left(x_{6i} - \overline{x_6}\right) + \beta_7\left(x_{7i} - \overline{x_7}\right) + \varepsilon \tag{1}$$

where $Y_i$ is the course level DFW grade, $\mu$ is the grand mean DFW grade, $\tau_i$ reflects the effect of PLTL semester on DFW grade, the term $\beta_1\left(x_{1i} - \overline{x_1}\right)$ indicates the effect of Spring vs. Fall semester, $\beta_2\left(x_{2i} - \overline{x_2}\right)$ reflects the effect of percentage of underrepresented students enrolled, $\beta_3\left(x_{3i} - \overline{x_3}\right)$ reflects the effect of average SAT of enrolled students, $\beta_4\left(x_{4i} - \overline{x_4}\right)$ indicates the effect of percentage of female students, $\beta_5\left(x_{5i} - \overline{x_5}\right)$ indicates the effect of percentage

of first-generation students, $\beta_6\left(x_{6i} - \overline{x_6}\right)$ indicates the effect of percentage of Pell students, $\beta_7\left(x_{7i} - \overline{x_7}\right)$ indicates the effect of placement tests administered and ε is the random residual error. All terms like $\left(x_i - \overline{x}\right)$ terms indicate the difference between a specific class and the average of all classes on that specific variable. In each case, $x_i$ represents an average of each class for a particular parameter, such as a semester or percentage of female students, whereas $\overline{x}$ represents the grand mean of all students in all semesters in that covariate.

**Random intercept analysis: Students and class level.** The third analysis used a random intercept model to estimate the impact of engagement in PLTL on course performance net the effect of both course and student level effects [39]. Random intercept models are a special case of hierarchical linear models in which the intercept varies based on the characteristics of a group in which individuals are nested [40]. In the case of this PLTL analysis, characteristics of the course enable deviations of the intercept in a student-level model, yielding a more accurate estimate of the effect of PLTL on student performance.

In this analysis, student-level grade data was retrieved from the university's Learning Management System for thirteen semesters (Fall 2002, Fall 2003, Fall 2004, Spring 2005, Fall 2005, Spring 2006, Fall 2016, Spring 2017, Fall 2017, Spring 2018, Fall 2018, Spring 2019, and Fall 2019). These data included the number of points earned on each assignment and scores on all tests. In each of these terms, students earn a PLTL score each week that depends on student preparation for the meeting and student engagement during the current PLTL session. Typically, 45–55% of students receive full credit for engagement in the sessions, distinguishing it from simple attendance data. Students in each term could earn a maximum of 180 PLTL points out of 1000 course points. The outcome (dependent) variable for these analyses was students' combined scores on the last two in-term tests (120 points each), representing 24% of the total course score. The last two tests (Exam 3 plus Exam 4 scores are denoted hereafter by "Ex$_{3+4}$") were selected as most representative of mature engagement with PLTL and the accumulated advanced content assessment. Independent of the random intercept analysis, we have also analyzed the impact of internal metrics on the URM/non-URM performance gap, considering assessment points and the total performance points (total points less the PLTL contribution).

Student GPA data was obtained from the university student information system and merged with student grade data for this analysis. Incoming GPA included high school GPA for first-time beginners who had transferred fewer than 15 credit hours into the university. For transfer students (who had completed at least 15 credit hours at another institution), GPA at their previous higher education institution was used. Additional course-level categorical variables representing "Fall vs Spring" and more recent terms "AY 2016–2018 and Fall 2019". The model depicting random intercept model is depicted, representing the student level and class level, respectively, in Eqs. 2 and 3.

$$Y_{ij} = \beta_{0j} + \beta_{1j}x_{ij} + \beta_{2j}x_{ij} + \varepsilon_{ij}, \tag{2}$$

where $Y_{ij}$ is the Ex$_{3+4}$ score, $\beta_{0j}$ is the intercept, $\beta_{1j}x_{ij}$ provides the centered PLTL engagement score, $\beta_{2j}x_{ij}$ defines the centered incoming high school or transfer GPA, and $\varepsilon_{ij}$ defines the random error, and

$$\beta_{0j} = U_{0j} + \delta_{01}x_j + \delta_{02}x_j \tag{3}$$

where $U_{0j}$ is the grand mean, $\delta_{01}x_j$ defines the course semester (Spring = 1, Fall = 0) and $\delta_{02}x_j$ defines the course was recent (AY 2016–2018 and Fall 2019 = 1, before Fall 2016 = 0).

For the random intercept analysis, at Level 1 (the student level, Eq. 2), we included the PLTL engagement scores and the incoming high school/transfer GPA as well as the random

intercept. Both these variables were centered on the mean to ease the interpretation of coefficients. At Level 2 (the course level, Eq. 3), we included a fixed overall intercept, the effect for course term (Spring or Fall), and the effect for recent course enrollments. These effects were included to account for significant differences in DFW grades between Fall and Spring terms as well as between recent and earlier terms.

**Ordinary least-square (OLS) demographic analysis.** The final analyses included separate ordinary least-square (OLS regression) analyses for only underrepresented students (self-designated African American/Black, Hispanic/Latine, Native American, Native Hawaiian/ Pacific Islander, or "two or more races") and students receiving a Pell Grant. These analyses were conducted to determine the differential effects of PLTL on underrepresented and Pell students. The data for these analyses were the same as for the random intercept model, with additional data merged with data from the student information system (race, ethnicity, and receipt of a Pell Grant). As with the random intercept model, the outcome variable for these analyses was the combined score $Ex_{3+4}$. The predictor (independent) variables included PLTL engagement score, incoming GPA (high school GPA for new beginners and transfer GPA for transfer students), term in which the course was held (Fall or Spring), and recency of the course (AY 2016–2018 and Fall 2019). Only students with a value for all variables were included in the model. Students who dropped the course before completion of Exam 3 and/ or Exam 4 were excluded. Students without an incoming GPA (either from high school or transfer) were also excluded. For students with both high school and transfer GPA, transfer GPA was used in the model. The full models for Pell and URM students are depicted in Eqs. 4 and 5, respectively:

$$Y = \beta_0 + \beta_1 x_1 + \beta_2 x_2 + \beta_3 x_3 + \beta_4 x_4 + \beta_5 x_5 + \varepsilon \ldots, \tag{4}$$

where Y is the $Ex_{3+4}$ score, $\beta_0$ is the intercept, the term $\beta_1 x_1$ represents the PLTL engagement score, $\beta_2 x_2$ represents the incoming GPA, $\beta_3 x_3$ is the receipt of a Pell grant, $\beta_4 x_4$ represents the course semester, $\beta_5 x_5$ represents the course was recent, and $\varepsilon$ is the random error. For the URM students,

$$Y = \beta_0 + \beta_1' x_1 + \beta_2' x_2 + \beta_3' x_3 + \beta_4' x_4 + \beta_5' x_5 + \varepsilon \ldots, \tag{5}$$

where Y is the $Ex_{3+4}$ score, $\beta_1' x_1$ represents the PLTL engagement score, $\beta_2' x_2$ represents incoming GPA, $\beta_3' x_3$ represents the URM student, $\beta_4' x_4$ represents the course term, $\beta_5' x_5$ represents the course was recent, and $\varepsilon$ represents random error.

The ANCOVA, random intercept, and OLS regression analyses were conducted using the SAS software (v. 9.4) package. Because data for random intercept models can yield inaccurate or confusing $R^2$ values, separate values for each level (class and student) were calculated using a SAS macro developed for constructing fit statistics in multilevel modeling [39]. Snijders and Bosker [40] indicated that a decrease in $R^2$ following the addition of fixed effects in level 2 could signal model misspecification. As such, the $R^2$ values have been used to determine if re-specification or the inclusion of additional variables is necessary.

## Results

### Course level analysis

The descriptive statistics of the 1996–2019 data used in the ANCOVA analyses in this study are placed in Supporting Information S1 Table. The %DFW and %ABC from AY 1996–1997 to the Fall 2019 are shown in Fig 1 with annotation of changing placement methodology used in this course. Spring 2020 and later course data were not included because of the

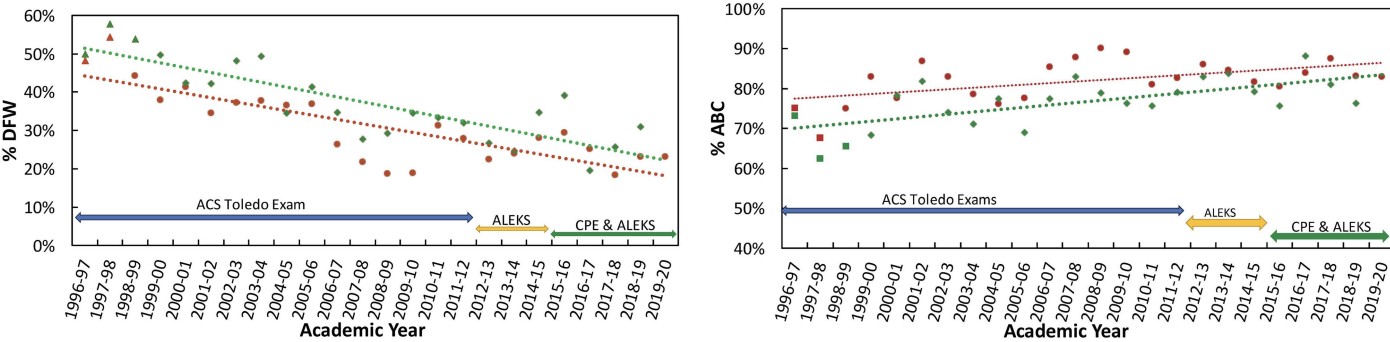

**Fig 1. Change in student retention (%DFW grades) and student success (%ABC grades) as a function of the academic year.** Left panel: The DFW rates in the Fall (red circle) and Spring (green diamond) semesters. Right panel: The %ABC in the Fall (red circle) and Spring (green diamond) semesters. Different placement methodologies are shown with double-headed arrows, such as the ACS Toledo Exam, ALEKS, and Chemistry Placement Exam with ALEKS. The dotted lines represent the linear regression for each data set, red for Fall and green for Spring semesters; the equations are %DFW = −1.13 × year + 45.33 with $R^2$ = 0.6672 for the Fall semesters and %DFW = −1.27 × year + 52.69 with $R^2$ = 0.6969 for the Spring semester. The equations are %ABC = 0.39 × year + 77.06 with $R^2$ = 0.2747 for the Fall semesters %ABC = 0.58 × year + 69.46 with $R^2$ = 0.4109 for Spring semester. In semesters where PLTL was not used are shown with filled squares.

interruptions due to the pandemic. The data were fitted using a linear regression model, which showed a decrease (a negative slope) in the %DFW over the years. The slopes ($\beta_1$) of the lines are −1.13 and −1.27, respectively, for the fall and spring semesters, indicating a slight difference between them. Similar plots of the %ABC also show an increase, although the slopes of the regression line are smaller.

Results from the ANCOVA analysis of PLTL on %DFW and related course-level covariates are described in Supporting Information S2 Table. The Type III Sum of Squares (SS III) displays the effect of each variable and net the effect on the other variables in the model. Degrees of Freedom (DF) are the extent to which a predictor variable or covariate is allowed to vary. Dividing SS III by DF will yield the Mean Square (MS), the variance. The F ratio (F), the ratio of the between-groups variance and the variance within groups (error), is used for the statistical significance test. Two effect size calculations for ANCOVA calculations, partial eta-squared ($\eta^2$) and partial omega-squared ($\omega^2$), were included to provide estimates of the magnitude of the observed effect. While $\eta^2$ is commonly reported in educational research, $\omega^2$ is less likely to be biased with small samples [41–43]. The effect size, a standardized measure across different studies, allows for an additional probe of statistical significance beyond the $p$-value alone. For example, $\eta^2$ (or $\omega^2$) values indicate that the effect of the SAT score is nearly twice that of the PLTL value (Supporting information S2 Table). The partial $\eta^2$ is the fraction of its SS to the sum of its SS III plus that of the error (or unexplained) sum.

The full model was statistically significant at $p \le 0.01$ [F(1,75) = 15.47)]. Terms in which PLTL was integrated as a part of the curriculum had a significant impact on DFW grade, net the effect of term, percentage of URM, female, first-generation, and Pell students combined SAT scores, and placement test administered [F(1,75) = 11.93]. Effect size estimates ($\eta^2$ = 0.137; $\omega^2$ = 0.114) both suggested a medium or more significant effect. The average SAT score [F(1,75) = 24.96] was the only variable with a significant net effect over the other variables. The effect size for the average course SAT score was large, with $\eta^2$ = 0.250 and $\omega^2$ = 0.220.

The marginal means from the ANCOVA analysis are included in Supporting information S3 Table. The means display the approximate mean %DFW for PLTL and non-PLTL sections, assuming the effect of the covariates was equal across groups. Even after adjusting for the covariates, the %DFW for course sections in which PLTL was part of the curriculum was

significantly lower (31.3%) than those without PLTL (47.2%). This suggests the effect of PLTL is consistent even when considering differences between class sections.

## In-course analyses

To establish the efficacy of different learning interventions, we have examined the correlations between different in-class assessments to establish their level of impact using the *Random Intercept Model*. With respect to the PLTL intervention, we investigated the correlations between the engagement scores in the PLTL section to exams and course points to assess potential impact. The points earned in PLTL sections reflect more engagement in the active learning process and less on the accuracy of student work in the session. In addition, the exams covered the major topical materials and were comparable to the coverage on exams throughout the period of study. Given changes in textbooks over time, language changes reflect contemporary usage. Four in-class exams are given, followed by a comprehensive final exam. Exam modality also changed over this period with a progression to online exams in proctored environments. Furthermore, students had flexibility in choosing the time to take individual exams over the full period.

Given the importance of course readiness, prior learning is reflected more in the earlier assessments than in later ones, and one expects a correlation between later exam scores and PLTL scores. Indeed, such a correlation reflects an increasing connection over the progression through the term. To quantify the early-late connection, we determined the correlations of total PLTL scores to the sum of the first two exam ($Ex_{1+2}$) scores and the last two exam ($Ex_{3+4}$) scores. Indeed, in most cases, we observed an increase in correlation between early vs. late exam performance and the PLTL scores. Four-year cohorts were used to average practices over adjacent terms. We conducted a detailed analysis by investigating an early PLTL cohort (Fall only, AY 2002–2005) and a late cohort (Fall only, AY 2016–2019), contrasting changes in the cohorts. The average $Ex_{3+4}$ scores between the two cohorts were also statistically significant. We believe the PLTL impact on test performance continues to improve as the actual program implementation becomes more efficacious. The means, standard deviations, and percentages for each variable included in the random intercept model and variables included in the ordinary least square (OLS regression) models are in Supporting information S4 Table. Here, we focus on selected covariates impacting student success in exams. For example, HS GPA addresses an academic success attribute, while the others reflect demographics or class modality. Only students with values for each variable are included below.

The results for the random intercept model are included in Supporting information S5 Table. The intercept represents the mean of $Ex_{3+4}$ for those included in the analysis. The $R^2$ for Level 1 was 0.3005, while the $R^2$ for Level 2 was 0.7496. The results suggest that the model appropriately fits the specified data. Four of the five effects in the full model were statistically significant net the effect of the other variables in the model at the $p \leq 0.01$ level. At the student level, PLTL was statistically significant in the net effect of the other predictor variables. Specifically, a one-point increase over the centered score in PLTL is associated with a 0.56-point increase in the predicted $Ex_{3+4}$ score. Incoming GPA and SAT were also statistically significant net the effect of the other predictor variables.

At the class level, courses taken more recently (Fall terms, 2016 - 2019) were associated with a 13.81 increase in $Ex_{3+4}$ score net the effect of the other predictor variables. This result suggests that students perform better on these tests in recent years than in the earlier years of PLTL implementation. In Supporting information S5 Table, the full random intercept model described in Eqs 2 and 3, where the centered PLTL engagement score is 0.56, the centered incoming GPA is 15.36, the centered SAT is 0.82, the fall/spring term differential is −2.45, and

the recent term effect + 13.81. The difference between fall and spring performances is large but not significant.

The results of the OLS regression using only students identifying as URM are listed in Supporting information S6 Table. As with the random intercept model, PLTL engagement, incoming GPA, and participation in a recent course were all significantly and positively related to the $Ex_{3+4}$ score. Receipt of a Pell grant was also significantly and negatively associated with $Ex_{3+4}$ score. The full model is shown in Eq. 5, and the coefficients are 0.61 for the PLTL engagement score, 3.43 for the incoming GPA, −6.85 for the Pell recipient, −5.62 for the course term, and 24.91 for the course was recent. The intercept is 133.44 depicted in Supporting information S6 Table. For PLTL, a one-point increase in PLTL engagement was associated with a 0.61 point increase in the $Ex_{3+4}$ score, net the effect of incoming GPA, receipt of a Pell grant, course term, and recency of course.

In Supporting information S7 Table, we tabulated the results of the OLS regression on the $Ex_{3+4}$ score for only those students receiving a Pell Grant. Here, PLTL engagement, incoming GPA, and recent course were statistically significant net the effect of the other variables in the model. Being an underrepresented student also had a statistically significant and negative effect. For Pell recipients, a one-point increase in PLTL engagement score was associated with a 0.57-point increase in the net $Ex_{3+4}$ score.

## Discussion

Our overall data in the 24-year analysis contains class-level trends in improvement in standard academic metrics, especially *student completion* (measured by %DFW in Fig 1) and *student success* (measured by %ABC in Fig 1). The grades in the figures are raw values showing both pre-PLTL and PLTL data. The linear regression lines indicate an overall improvement in both grades but contain term-to-term variability. Placement exams have been used throughout the entire window to assist students in identifying the most appropriate enrollment choices. We have also found an improving trend in the pre-enrollment data, such as SAT scores and high school/transfer GPAs. Clearly, the data show an improvement in %DFW and %ABC over the years. Before interpreting the improvement in detail, the prospect of a grade inflation effect is examined.

### Grade inflation analysis: Cohort comparisons

One important question is whether the standards were lowered deliberately or unknowingly, assessing students as successful or increasing completion, i.e., *grade inflation* [44,45]. In general, there are two types of grade inflation: *static* grade inflation, i.e., disconnection from grade assignment from content mastery within individual sections, and differential grade inflation, i.e., the relationship between grades based on different term enrollments. To understand the grade inflation, if any, we evaluated data initially described in Supporting information S8 Table, that could inform the likeliness of grade inflation occurring. This evaluation aimed to assess any dynamic grade inflation arising from drifting standards in grade assignments over time. To address this question, we examined whether the increases in course success and/or completion metrics correlate to performance on the standardized internal assessments. An increase in median score value is often viewed as an indicator of potential grade inflation. Therefore, it is important to ensure that such an increase reflects the improvement in content expertise evaluated by assessments.

The fall sections from the two 4-year cohorts discussed earlier were analyzed to evaluate the *internal* course metrics and their impact on completion and success metrics. Since the output metrics involve grades A through F and/or W, factors potentially altering these assignments

were controlled. Sections in the cohorts were restricted to fall offerings; the difference between fall and spring performances is large but not significant. Relevant pre-enrollment, in-class final exams, and course GPA statistics of each section are included in Supporting information S9 Table.

The courses in these two cohorts have been graded on exam performance and additional metrics, such as PLTL participation scores, homework, and class response systems (e.g., clickers). The course content has been generally consistent in the two cohorts. Following the typology of grade inflation by Tyner and Gershenson [45], all individual semester sections shared a common syllabus, used the same grading scale for final letter grade assignments (control for *static* grade inflation), and used identical midterm/final examinations within all sections/instructors (controlling for *differential* grade inflation) in any given term [46]. Midterm exams have a common 2-hour maximum test time using exams constructed for a 70-minute period to improve the validity of testing assessments [47]. Students had the option to choose any available time period over a three-day testing window to take an exam to accommodate student preferences and availability. Grade thresholds were not curved but were based on a common scale to minimize inflationary tendencies and to improve equity in the course grade assignments [48]. While there were slight variations in assignments of plus/minus grades, more recent terms used slightly higher breakpoints for letter grade assignments than the earlier cohort (about +1% for A and B and +3% for C). The four internal exams largely covered similar content in both cohorts. An internal, semi-standardized final exam with comparable course content coverage has been used during the full period of this study. Initially, the final exam was administered in written form for all sections within a term, but in later years, it was converted to an online version in a proctored environment.

While internal median total examination metrics improved by ~16% between the early and late cohorts, the course average GPA only increased by 8%, implying a much smaller response than might be expected. Inflation would be implied if the average course GPA increased by more than 16%. Even the %ABC increased by only 7% for all students completing the course. The grade median increased from a C+ (67.8% total points earned) to a B− (78.6% total points earned), netting a 10.8% change, still less than internal examination metrics. In conclusion, students' improved performance demonstrating content understanding exceeded the grades awarded in the course, implying an absence of grade inflation.

## PLTL's impact on course output variables: %DFW and %ABC

The most notable outcome from the analysis of the full data is the continuing improvement in the %DFW, from a pre-PLTL marginal level of 47% to an overall PLTL level of 31%, in which the SAT score of the incoming students is an important contributor (Supporting information S2 and S3 Tables). It is clear in the latter part of the period that the %DFW continues to decrease, as shown in Supporting information S5 Table (from the significant β value of 20.04 for enrollments "AY 2016–2018 and Fall 2019") and Fig 1. In terms of student success, the %ABC indicates the fraction of course completions that lead to credits applying to most degree requirements. The overall %ABC increased from 69% pre-PLTL to 82% with PLTL. This result is believed to follow from a general improvement in the performance of the entire ABC group.

## Students course progression and active learning

Since the early content relies more on pre-requisite material, students are comfortable re-learning that material, but as the term progresses, the content becomes increasingly less familiar, underscoring the value of the PLTL intervention. To evaluate the impact of this

intervention Exam scores were examined in relation to PLTL scores. For example, Exam 4 scores from Fall 2018 vs PLTL engagement scores were fitted with linear regression (Supporting information S1 Fig). The robustness of the regression analysis was evidenced by a lack of heteroscedasticity in the scatter plot residuals. Fig 2 shows the changes in the regression line slopes (β) for each exam vs. engagement score within the early/late cohorts. The figure demonstrates that the slope increases for the latter exams within each cohort, suggesting that students perform better as they get more experience in the PLTL sessions, relying less on prior learning. Thus, the largest slopes were generally observed in later exams during the term, while the smallest slopes were in earlier exams. We surmise that this progressive increase in performance has a common thread of greater engagement with the PLTL process, and thus, PLTL serves as one of the drivers for enhanced exam performance. Between the early and late cohorts, the overall numbers are higher in the late cohorts. We posit that as the PLTL model became better tuned over time, it enhanced students' learning outcomes.

## URM/non-URM student outcomes

Given the importance of overall trends in the URM/non-URM gap analysis reported by others [11,34–36], we explored the implications of PLTL improvements on the gap. Overall, %ABC and %DFW metrics were disaggregated by URM status (Supporting information S11 Table). Fig 3 shows the behaviors in the progression of %DFW/%ABC for each section during the study period, starting with Spring 1997. Data from 1997 were included as an anchoring point. The plots show the student success (i.e., A, B, and C grades) and the student completion (D, F, and W grades) have progressively improved over the period of the study, and the data were fit to linear regression lines. The slope of the lines can be taken as a quantitative marker for the change over the years indicated. The enhancement in student success rate is higher in the spring semester compared to the fall semester for both URM and non-URM students: 1.5 times higher for URM students, and 1.4 times higher for non-URM students. However, there was no observable difference in the student completion rate between the spring and fall semesters at 1.1 for both for URM and non-URM students. The success rate improvements of

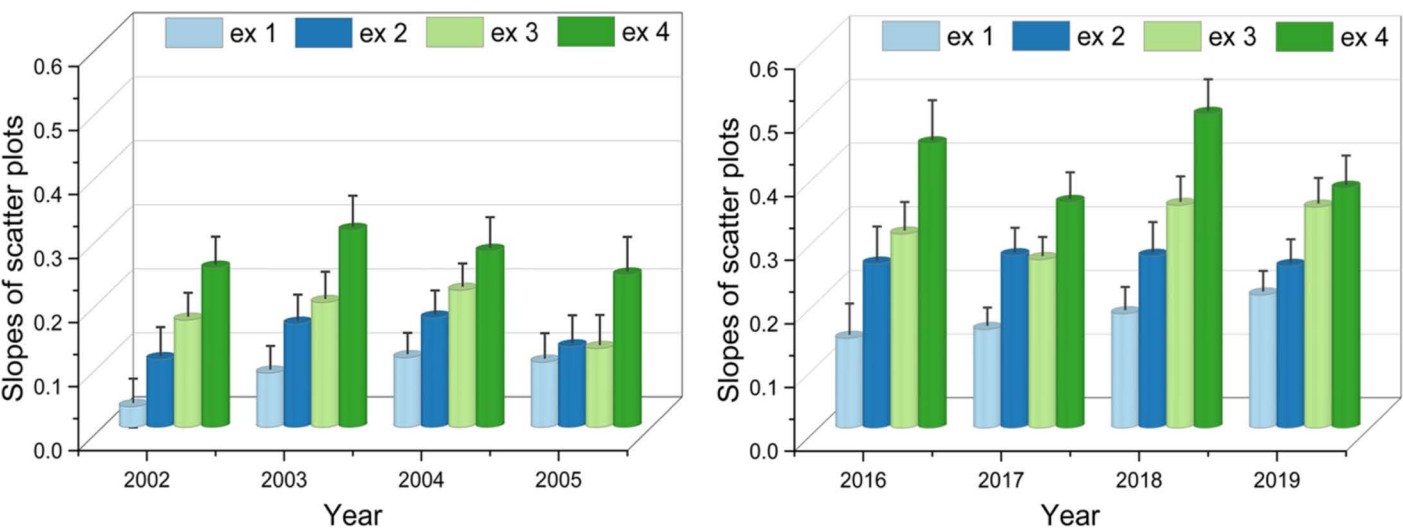

**Fig 2. Progression of exam scores in each year within the cohorts.** Bar diagrams displaying the change in the slopes of the scatter plots of exams taken by early (left panel) and late (right panel) cohorts. In each case, the errors in the slopes are also included. The diagram demonstrates that continued engagement in PLTL improved exam performance.

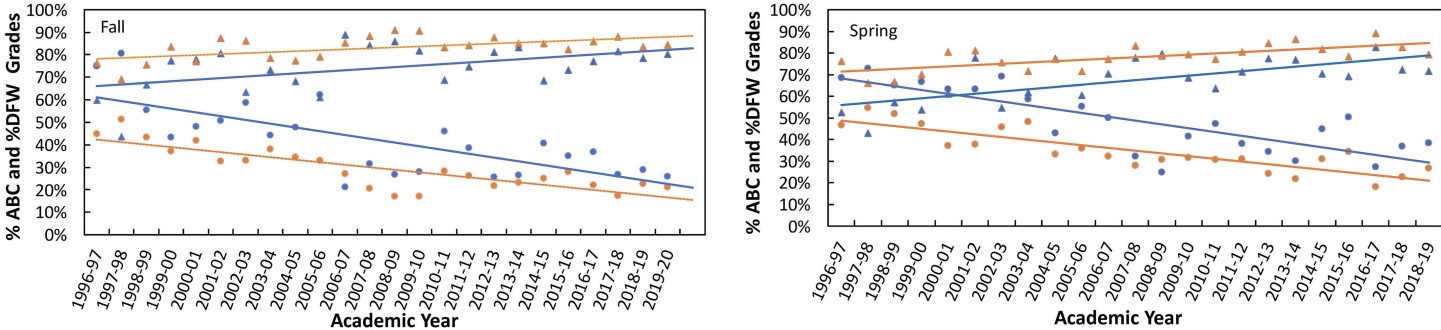

**Fig 3. Progression of %DFW and %ABC grades separated by URM and non-URM students.** The %ABC (filled triangle) and %DFW (filled circles) grades for each semester for URM (blue) and non-URM (orange) students. The data were fitted with a linear regression model in each category. For the Spring semesters, %ABC of URM students = 0.0104 × year + 0.5494 with $R^2$ = 0.4604; %ABC of non-URM students = 0.0060 × year + 0.708 with $R^2$ = 0.4832; %DFW of URM students = −0.0177 × year + 0.6998 with $R^2$ = 0.6489; %DFW of non-URM students = −0.0126 × year + 0.05002 with $R^2$ = 0.7344. For the Fall semesters, %ABC grades of URM students = 0.0070 × year + 65.42 with $R^2$ = 0.2334; %ABC grades of non-URM students = 0.0043 × year + 0.7772 with $R^2$ = 0.3222; %DFW of URM students = −0.0167 × year + 0.6277 with $R^2$ = 0.5513; %DFW of non-URM students = −0.0112 × year + 0.4344 with $R^2$ = 0.6886.

the URM to non-URM students are 1.6 and 1.7 for the Fall and Spring semesters, respectively. The completion rate improvements of the URM to non-URM students are 1.5 and 1.4 for the Fall and Spring semesters, respectively. Thus, URM students performed better in the student success metric than their non-URM counterparts. These results reinforce that URM students are impacted more by PLTL strategies than non-URM students as measured by %DFW and %ABC [49].

The impact of PLTL on student demographics was further explored by analyzing the cohorts' data as described in Table 1 and Supporting information S9–S11 Tables. The data in these tables further underscore the improvement in the URM student output metrics compared to non-URM students. Two internal metrics, performance on the final exam and total performance points (excludes engagement points from PLTL sections), showed greater changes in URM performance (21% and 26%, respectively). This enhancement in performance is manifested in the striking improvements in the average course grades of URM vs. non-URM, 19% to 9% (Supporting information S10 Table).

The changes in URM/non-URM gaps in course GPA averages (Supporting information S2 Fig) show large gap improvements progressing from pre-PLTL terms to the most recent cohort. The pre-PLTL gap of 0.53 GPA, shrinks to a modest 0.2 difference in that late cohort. The %ABC-based gap over the same period shrunk by 70%. The %DFW progression from pre-PLTL to the early cohort shows the first transition of 38%, followed by a reduction from early to late of 52% and an overall net reduction of 71%.

A similar URM/non-URM analysis in an introductory biology course reported substantial improvements in the %DFW and gap differences by comparing PLTL and non-PLTL sections [49]. Our gap analysis of the URM/non-URM students also shows a reduction in multiple course assessments between the early and late cohorts (Table 1). The gap improvements in %ABC and %DFW are 56% and 52%, respectively (Table 2).

Data in Table 1 also implies a more complex situation with the gender (self-reported binary gender identity) gaps. For example, the average course grade and the average $Ex_{3+4}$ scores are within each standard error of each measurement. In the final exam category, gender scores were within the standard errors of the net value. In the performance points category, an existing statistically evident gap was present; however, this gap diminished in the late cohort to a non-discernable gap. In all cases, as with the URM situation, major reductions in gaps continued to occur.

**Table 1. URM/non-URM and Gender performance gaps in selected metrics.**

| Performance Metrics | non-URM/URM Gap | | | Gender (M-F) Gap | | |
|---|---|---|---|---|---|---|
| | Early Cohort | Late Cohort | % Change* | Early Cohort | Late Cohort | % Change* |
| **HS GPA, 4-pt scale, raw[†]** | 0.12 | 0.08 | −30%[‡] | −0.21[‡] | −0.14[‡] | −30%[‡] |
| **Total SAT scores, raw[†]** | 85 | 94 | 11%[‡] | 39[‡] | 29[‡] | −25%[‡] |
| **%Final Exam scores** | 6.21% | 4.84% | −22%[‡] | 2.13%[‡] | 2.00%[‡] | −6%[‡] |
| **%Ex$_{3+4}$ scores** | 4.26% | 3.23% | −24%[‡] | 0.89%[§] | −0.09%[§] | −110% |
| **%Performance Points[‖]** | 5.16% | 4.04% | −22%[‡] | 1.16%[‡] | 0.14%[§] | −88% |
| **Course GPA Gap, 4-pt scale** | 0.391 | 0.230 | −41%[¶] | 0.04[§] | −0.05[§] | −226% |
| **Achievement score gap** | 18.71% | 8.92% | −52%[¶] | 2.48% | 0.39% | −123% |

* %Change, $(X_{LC} − X_{EC})/X_{EC}$.

[†] "Raw" is the numerical difference in grades/scores in gap values.

[‡] Non-negligible gap.

[§] No net gap, with values within the standard error range.

[‖] *Performance points* are total course points minus PLTL engagement points.

[¶] The *achievement score* is defined as "100 – %DFW". Interestingly, a 33% reduction in the achievement gap in all STEM fields between pre- and post-PLTL has been reported. An average of 0.54 lower grade points for URM students than non-URM students [36].

**Table 2. URM/non-URM Gap changes in student success and completion metrics.**

| *Fall Cohort gap (Cohort size)* | Completion: %DFW change | Success: %ABC change |
|---|---|---|
| **Pre-PLTL Cohort Gap (588)** | 30.08%* | 21.20%* |
| **Early PLTL Cohort Gap (1268)** | 18.71%* | 14.14%* |
| **Late PLTL Cohort Gap (2079)** | 8.92%* | 6.25%* |
| **% Change in Pre-PLTL/Early Cohort Gap** | −38%[†] | −33%[†] |
| **% Change in Early/Late Cohort Gap** | −52%[†] | −56%[†] |
| **% Change in Pre-PLTL/Late Cohort Gap** | −71.2%[†,‡] | −70.3%[†,‡] |

* Numerical difference, $X_{non-URM}-X_{URM}$.

[†] Fractional change, $(X_b − X_a)/X_a$.

[‡] Literature reports a 45% reduction in the gap for passing rates comparing pre-PLTL with PLTL sections in STEM courses [36] and an 80% DFW grade reduction in biology courses [49].

We have also examined URM/non-URM %ABC and %DFW disaggregated by gender. While the initial analysis controlling for URM and gender effects is reported in Table 2, the size effects for URM and gender both implied a small to medium effect. In Fig 4, we show the overall %ABC and %DFW for non-URM and URM populations. In the non-URM case, gender differences are not consistent over the term of the study data. Female students performed better in about ⅔ of the cases than males in both %ABC and %DFW comparisons. In the URM cases, the outcomes are reversed, with males having about ⅔ of the more favorable outcomes in both grades.

## Conclusions

In this work, we have evaluated the data of 19,911 students who enrolled in our first semester of the 2-semester General Chemistry sequence over a 24-year period, where 18,818 students enrolled with the PLTL intervention. Data collected between 1996–1998 using pre-PLTL data from five semesters (1100 students) served as a control. We investigated different

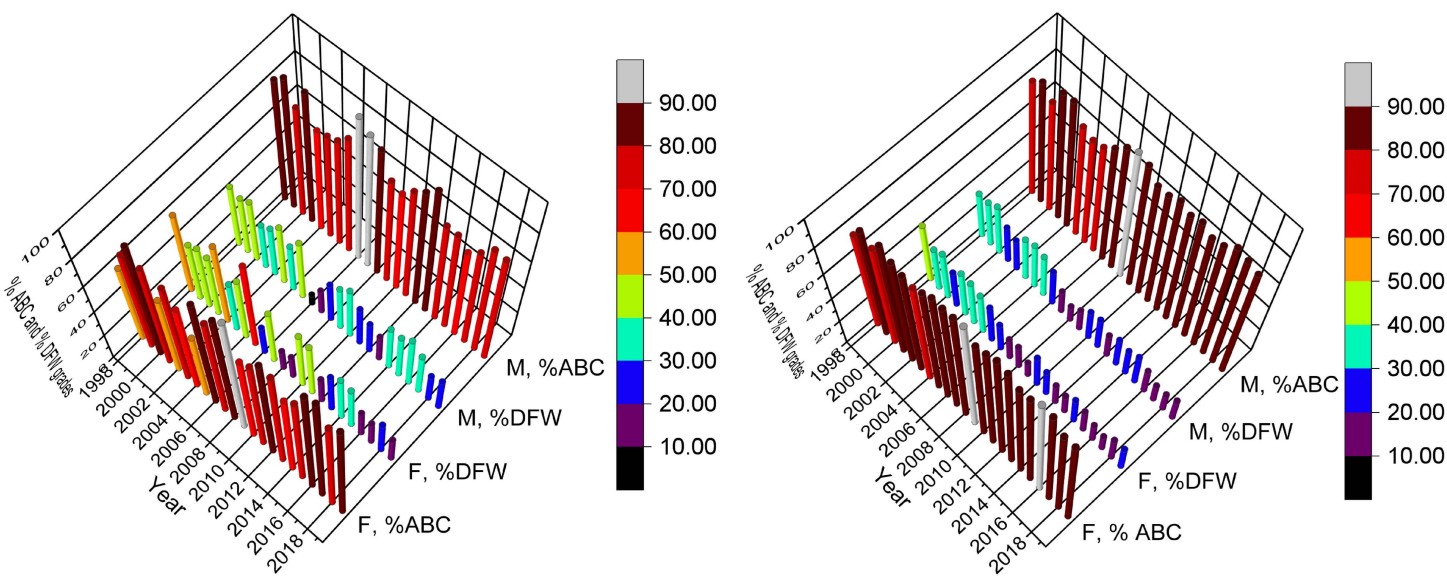

**Fig 4. Student success and retention separated by gender and URM status.** Comparison of %ABC and %DFW disaggregated by self-reported binary gender (female and male) for non-URM (left panel) and URM (right panel) students.

demographic and scholastic factors that may contribute to the success and completion of General Chemistry. The data were analyzed using ANCOVA and the HLM Random Intercept Model. The following overall conclusions address the original questions asked of our intervention on success/completion, efficacy in various student demographics, internal assessment validity, and the use of engagement measures:

- **Overall completion metric, %DFW**. In the largest-ever longitudinal study of PLTL/non-PLTL classes, 19,000 student data demonstrated that *PLTL and SAT had a statistically significant and large effect size on the overall completion metric* (%DFW). In addition, while the First Generation, URM and gender covariates were not statistically significant contributors, they had a small to medium effect size on the completion metric. Accompanying the completion rate, the student success rate (%ABC) also shows improvement.

- **Pre-PLTL, early cohort (2002–05), and late cohort (2016–19) gap impact**. *URM/non-URM performance gaps between early and late cohorts* have been reduced by greater than 50% in course metrics and more than 20% in final exam metrics ($p < 0.01$). Pre-PLTL to Late cohort performance gaps were reduced by 70%. Moreover, *gender gaps have virtually disappeared in* several metrics, including late *internal exam scores and overall average class GPA*.

- **Sustained attention to best-practice pedagogies yields continuing improvements in course metrics**. Intraclass analysis reveals a significant impact of PLTL; however, sustained monitoring is essential to greater longitudinal improvements. While the improvements are independent of instructors, the continued professional development of instructors is important [50]. In addition, our analysis shows students' achievements exceeded those expected from grade inflation.

- **Engagement scores impact exam performance and success/completion metrics in the cohort analysis.** Engagement points based on measures of participation and pre/post involvement correlate favorably with exam scores and earned course grades.

- **Linear mixed models analysis of intraclass performance has identified the statistically significant contributors to in-course performance metrics**. PLTL engagement scores, HS/Transfer GPAs, SAT scores, and term enrollment recency are all significant covariates for predicting late exam performance (at the $p < .01$ level). In addition, PLTL engagement points contribute to course Success/Completion metrics with statistical significance.

While our application has been in general chemistry, we envision that this approach could be adapted to other STEM gateway courses through appropriate workshop materials and leader training. Thus, the PLTL can, not only help reduce the overall loss of students from STEM disciplines, but also reduce the achievement gap between URM and non-URM students, one of the most important challenges in higher education today.

## Supporting information

**S1 Fig. A representative scatter plot of Exam 4 score distribution vs. PLTL engagement score, Fall 2018.** The residuals plot shows that heteroscedasticity is not apparent and implies the fitting satisfies validity criteria. The equation for the linear regression line (red) is shown. In the PLTL sections, leaders assess students' engagement with content preparation, problem-solving, participation, and post-section worksheets.
(TIFF)

**S2 Fig. Analysis of the non-URM/URM average course GPA gaps shown as blue bars for early (Fall 2002–2005) and late (Fall 2016–2019) cohorts.** The HS GPA shown in orange bars indicates the gaps in those averages in the two cohort periods. There is insufficient data for pre-PLTL comparisons with HS-GPA.
(TIFF)

**S3 Fig. STEM degree completion by academic year.** The figure shows the fraction of students with initial STEM interest that successfully complete a STEM degree program (in a six-year window). The PLTL program started in Fall, 1998, so the first impact of PLTL would likely be starting in AY 2002–03 and later.
(TIFF)

**S1 Table. Course-level statistics (1996–2019).**
(DOCX)

**S2 Table. Effect of PLTL on course-level DFW grades.**
(DOCX)

**S3 Table. Marginal means for PLTL and non-PLTL course sections.**
(DOCX)

**S4 Table. Student and course level statistics for variables included in the random intercept model.**
(DOCX)

**S5 Table. Random intercept model: Effect of PLTL engagement on the $Ex_{3+4}$ score.**
(DOCX)

**S6 Table. OLS regression on $Ex_{3+4}$ score for URM students only.**
(DOCX)

**S7 Table. The OLS regression on the $Ex_{3+4}$ score for Pell recipients.**
(DOCX)

**S8 Table. Comparison metrics (Pre-enrollment, internal, and output) for early and late cohorts.**
(DOCX)

**S9 Table. Detailed comparative performance within cohorts.**
(DOCX)

**S10 Table. Comparative performances (Fall early vs. Fall late cohorts) of URM and non-URM student Final Exam scores, course grades, course DFW rates, and course ABC rates. Fall terms only.**
(DOCX)

**S11 Table. Fall early/late cohort output metrics: %ABC and %DFW with URM status.**
(DOCX)

## Acknowledgments

We thank Profs. Pratibha Varma-Nelson and Lyniesha Ward for helpful discussions. We also thank Dr. Danka Maric for carefully reading the manuscript and Dr. Nitai Giri for assistance in preparing some of the figures. We also thank Teresa Troke for her help with data collection.

## Author contributions

**Conceptualization:** Partha Basu.

**Data curation:** Steven Graunke.

**Formal analysis:** Partha Basu, David J. Malik, Steven Graunke.

**Investigation:** David J. Malik.

**Methodology:** Steven Graunke.

**Writing – original draft:** Partha Basu.

**Writing – review & editing:** Partha Basu, David J. Malik.

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
