## [Decision Letter · Decision Letter 0]

27 Dec 2024

PONE-D-24-42866A 24-year Longitudinal Study on a STEM Gateway General Chemistry Course and the Reduction of Achievement DisparitiesPLOS ONE

Dear Dr. Basu,

Thank you for submitting your manuscript to PLOS ONE. After careful consideration, we feel that it has merit but does not fully meet PLOS ONE’s publication criteria as it currently stands. Therefore, we invite you to submit a revised version of the manuscript that addresses the points raised during the review process. I also want to add my apologies for the delays with the manuscript.  It is becoming increasingly difficult to find qualified reviewers for manuscripts and for this  reason I have decided to proceed and ask for a revised manuscript since the reviewer's comments are quite positive and the requested changes quite minor.

We look forward to receiving your revised manuscript.

Kind regards,

Luís A. Nunes Amaral, Ph.D.

Academic Editor

PLOS ONE

Journal Requirements:

“There is no competing interest. “

Reviewers' comments:

Reviewer's Responses to Questions

**Comments to the Author**

1. Is the manuscript technically sound, and do the data support the conclusions?

Reviewer #1: Yes

2. Has the statistical analysis been performed appropriately and rigorously? 

Reviewer #1: Yes

3. Have the authors made all data underlying the findings in their manuscript fully available?

Reviewer #1: Yes

4. Is the manuscript presented in an intelligible fashion and written in standard English?

Reviewer #1: Yes

5. Review Comments to the Author

Reviewer #1: The authors have presented a 24 year longitudinal study on student performance in general chemistry and have assessed the role of Peer-Led Team Learning on completion rates amongst several groups of students. The article is well-written, generally clear, and presents a compelling case for the importance of their finds. I recommend two additional changes to improve the article and make it easier for future readers to comprehend and understand.

• Line 126: Please consider including selected references to refer curious reads to principles of PLTL. The text references this idea throughout, but in my opinion, it lacks a comprehensive description. This is acceptable for publication if appropriate resources are suggested to the reader. I believe additional citations would be useful to provide a starting point for chemical educators to follow-up on and adaopt. In contrast, I would also suggest the authors consider including at least one paragraph that focuses on the principles of PLTL in their own words and combine that with specific details of how this was implemented in their study.

• Line 192: This is the first time in the manuscript that %DFW and %ABC are referenced. As a general chemistry instructor who does not focus my research on chemical education these two terms were initially very confusing and appeared in the abstract and introduction. I recommend introducing their meaning earlier at their first instance, perhaps as early as the abstract.

Overall, the results are statistically rigorous and the conclusions are sound and informed. I recommend publication after minor revision.

6. PLOS authors have the option to publish the peer review history of their article (what does this mean? ). If published, this will include your full peer review and any attached files.

**Do you want your identity to be public for this peer review?** For information about this choice, including consent withdrawal, please see our Privacy Policy .

Reviewer #1: No

---

## [Author Response · Author response to Decision Letter 1]

2 Jan 2025

All comments from the reviewer and those of the editor have been addressed.

---

## [Editor Report · Decision Letter 1]

24 Jan 2025

A 24-year Longitudinal Study on a STEM Gateway General Chemistry Course and the Reduction of Achievement Disparities

PONE-D-24-42866R1

Dear Dr. Basu,

We’re pleased to inform you that your manuscript has been judged scientifically suitable for publication and will be formally accepted for publication once it meets all outstanding technical requirements.

Kind regards,

Luís A. Nunes Amaral, Ph.D.

Academic Editor

PLOS ONE
---

## [Editor Report · Acceptance letter]

PONE-D-24-42866R1

PLOS ONE

Dear Dr. Basu,

I'm pleased to inform you that your manuscript has been deemed suitable for publication in PLOS ONE. Congratulations! Your manuscript is now being handed over to our production team.

Kind regards,

on behalf of

Dr. Luís A. Nunes Amaral

Academic Editor

PLOS ONE